# Distinct Localization of Mature HGF from its Precursor Form in Developing and Repairing the Stomach

**DOI:** 10.3390/ijms20122955

**Published:** 2019-06-17

**Authors:** Nawaphat Jangphattananont, Hiroki Sato, Ryu Imamura, Katsuya Sakai, Yumi Terakado, Kazuhiro Murakami, Nick Barker, Hiroko Oshima, Masanobu Oshima, Junichi Takagi, Yukinari Kato, Seiji Yano, Kunio Matsumoto

**Affiliations:** 1Division of Tumor Dynamics and Regulation, Cancer Research Institute, Kanazawa University, Kakuma, Kanazawa 920-1192, Japan; nawaphatice@gmail.com (N.J.); hiroki.sato@staff.kanazawa-u.ac.jp (H.S.); imamura@staff.kanazawa-u.ac.jp (R.I.); k_sakai@staff.kanazawa-u.ac.jp (K.S.); 2WPI-Nano Life Science Institute (WPI-NanoLSI), Kanazawa University, Kakuma, Kanazawa 920-1192, Japan; hoshimam@staff.kanazawa-u.ac.jp (H.O.); oshimam@staff.kanazawa-u.ac.jp (M.O.); syano@staff.kanazawa-u.ac.jp (S.Y.); 3Division of Epithelial Stem Cell Biology, Cancer Research Institute, Kanazawa University, Kakuma, Kanazawa 920-1192, Japan; y.terakado@staff.kanazawa-u.ac.jp (Y.T.); kmurakami@staff.kanazawa-u.ac.jp (K.M.); nicholas.barker@imb.a-star.edu.sg (N.B.); 4A*STAR Institute of Medical Biology, Singapore 138648, Singapore; 5Division of Genetics, Cancer Research Institute, Kanazawa University, Kakuma, Kanazawa 920-1192, Japan; 6Laboratory of Protein Synthesis and Expression, Institute for Protein Research, Osaka University, Osaka 565-0871, Japan; takagi@protein.osaka-u.ac.jp; 7Department of Antibody Drug Development, Tohoku University Graduate School of Medicine/New Industry Creation Hatchery Center, Tohoku University, Sendai 980-8575, Japan; yukinarikato@med.tohoku.ac.jp; 8Division of Medical Oncology, Cancer Research Institute, Kanazawa University, Kakuma, Kanazawa 920-1192, Japan; 9Tumor Microenvironment Research Unit, Institute for Frontier Science Initiative, Kanazawa University, Kanazawa 920-1192, Japan

**Keywords:** HGF, MET receptor, regeneration, smooth muscle cell, stem cell

## Abstract

Hepatocyte growth factor (HGF) is secreted as an inactive single-chain HGF (scHGF); however, only proteolytically processed two-chain HGF (tcHGF) can activate the MET receptor. We investigated the localization of tcHGF and activated/phosphorylated MET (pMET) using a tcHGF-specific antibody. In day 16.5 mouse embryos, total HGF (scHGF + tcHGF) was mainly localized in smooth muscle cells close to, but separate from, MET-positive epithelial cells in endodermal organs, including the stomach. In the adult stomach, total HGF was localized in smooth muscle cells, and tcHGF was mainly localized in the glandular base region. Immunostaining for pMET and Lgr5-driven green fluorescent protein (GFP) indicated that pMET localization overlapped with Lgr5^+^ gastric stem cells. HGF promoted organoid formation similar to EGF, indicating the potential for HGF to promote the survival and growth of gastric stem cells. pMET and tcHGF localizations changed during regeneration following gastric injury. These results indicate that MET is constantly activated in gastric stem cells and that the localization of pMET differs from the primary localization of precursor HGF but has a close relationship to tcHGF. Our results suggest the importance of the microenvironmental generation of tcHGF in the regulation of development, regeneration, and stem cell behavior.

## 1. Introduction

Hepatocyte growth factor (HGF) and its receptor MET play roles in embryonic development and the repair of tissues following injury [1,2,3]. HGF is a fibroblast-derived morphogen that induces epithelial branching duct formation [4,5]. Scatter factor, a fibroblast-derived cell motility factor for epithelial cells, is identical to HGF [6]. In previous studies, the secretion of HGF was not noted in epithelial cells, but was in fibroblasts, vascular smooth muscle cells, and hematopoietic cells in culture [7,8]. Subsequent studies revealed the roles of HGF in the growth and morphogenesis of different types of epithelial cells and tissues as a mediator of epithelial–mesenchymal interactions [9,10,11,12,13].

Mice lacking HGF were found to exhibit embryonic lethality due to impaired development of the placenta and liver [14,15]. In addition, MET-deficient mice had no skeletal muscle in the limbs and diaphragm because of the impaired migration of myogenic precursor cells to the limb buds and diaphragm [16]. The characterization of conditional MET-deficient mice via analyses on a variety of cell types revealed that the HGF–MET pathway promotes tissue regeneration and protection, and suppresses the progression of chronic inflammation and fibrosis [3,17,18,19,20].

HGF is a glycosylated protein composed of α- and β-chains linked by a disulfide bridge. It is biosynthesized and secreted as single-chain HGF (scHGF); the cleavage of scHGF at the Arg494–Val495 generates two-chain HGF (tcHGF) [21,22]. scHGF is a biologically inactive precursor incapable of activating the MET receptor, while tcHGF is the only active molecular species that can activate it. Previous studies indicated that HGF exists as scHGF in normal tissues and that its processing into tcHGF occurs in injured tissues [23] and the tumor microenvironment [21,24]. Aberrant activation of the HGF–MET pathway, including ligand-dependent MET activation, is strongly associated with invasive growth, metastasis, and resistance to anticancer drugs [2,22].

These previous findings suggest that the localization of tcHGF is key to understanding how the HGF–MET pathway participates in tissue development, regeneration, and cancer progression. However, there are currently no molecular tools that specifically recognize tcHGF but not scHGF. Because the stomach is a typical organ composed of epithelial and mesenchymal cells, and stem cells play important roles in development and regeneration [25], analyses of the localization of sc/tcHGF and its association with MET activation in the stomach may provide a better understanding of how the HGF–MET pathway participates in development, regeneration, and the stem cell behavior of endoderm-derived organs. In a recent study, we obtained several anti-HGF monoclonal antibodies that recognize different epitopes of human HGF [26], one of which selectively recognizes human tcHGF. In this study, we thus investigated the localization of tcHGF and active MET in the developing and regenerating stomach, employing human tcHGF-specific monoclonal antibody and human HGF-knock-in mice.

## 2. Results

### 2.1. Specificity of Antibodies

Our previous study indicated that the t5A11 anti-human HGF monoclonal antibody recognizes both biologically inactive scHGF and active tcHGF, whereas the t8E4 monoclonal antibody exclusively recognizes tcHGF [26]. We first confirmed the cross-reactivity and specificity of the antibodies used in this study using western blot analysis and enzyme-linked immunosorbent assay (ELISA). Mouse HGF and human HGF (composed of scHGF and tcHGF) were subjected to SDS-PAGE and western blotting under non-reducing conditions (Figure 1 and Appendix A). A commercial polyclonal anti-mouse HGF antibody was reactive to mouse and human HGF (Figure 1A), while t5A11 and t8E4 recognized human HGF but not mouse HGF (Figure 1A and Appendix A). t5A11 was reactive to scHGF and tcHGF, whereas t8E4 selectively recognized tcHGF but not scHGF (Figure 1B). Thus, t5A11 and t8E4 antibodies were only reactive to human HGF, t8E4 was specific to tcHGF, and t5A11 was reactive to both scHGF and tcHGF.

### 2.2. HGF and MET Receptor Localizations in Human Knock-in Mice

Based on the specificity of t5A11 and t8E4 antibodies to human HGF, we decided to use human *HGF* knock-in (hHGF-ki) mice obtained from the Jackson Laboratory (Hgf^tm1.1(HGF)Aveo^Prkdc^scid^/J). In the hHGF-ki mice, both alleles of exons 3–6 of the endogenous murine *HGF* gene were replaced with a cDNA sequence encoding exons 2–18 of the human *HGF* gene. Human HGF was detectable but mouse HGF was not detectable in the plasma of hHGF-ki mice [27].

To confirm the compatibility in the expression and localization of HGF between wild-type C57BL/6 and hHGF-ki mice, immunohistochemical and immunofluorescence detection was performed using 16.5 days post-coitum mouse embryos from wild-type C57BL/6 and hHGF-ki mice (Figure 2 and Appendix A). In the developing stomach and intestine of wild-type C57BL/6 mice, HGF was distributed mainly in mesenchymal cells but faintly in epithelial cells. α-Smooth muscle actin (α-SMA) was expressed in smooth muscle cells and myofibroblasts. α-SMA staining indicated that HGF-positive cells were mainly smooth muscle cells and myofibroblasts in the sub-epithelial region. In hHGF-ki mice, HGF was mainly localized in smooth muscle cells, while it was weakly present in myofibroblasts in the sub-epithelial region and in epithelial cells. These results indicate that smooth muscle cells were the main cellular source of HGF and that hHGF-ki mice were an appropriate tool to investigate the localization of HGF.

By day 16.5, the embryos of hHGF-ki mice had already developed a variety of tissues/organs. HGF-positive cells were mainly smooth muscle cells of several organs including the esophagus, trachea, lung, stomach, intestine, and urinary bladder (Appendix A). The MET receptor was mainly localized in epithelial cells. Previous studies indicated that HGF regulates the growth and morphogenesis of different types of epithelial cells and tissues, mainly as a mesenchymal-derived paracrine factor [4,5,9,10,11,12,13]. Thus, these expression patterns of HGF and the MET receptor in developing tissues suggest that HGF and MET play roles in the development of several organs.

### 2.3. tcHGF and Phosphorylated MET Receptor in the Developing Stomach

To clarify the potential involvement of the HGF–MET pathway in the development of the stomach, the localization of total HGF and MET was analyzed using day 16.5 embryos (Figure 3 and Appendix A). α-SMA staining delineated a line of smooth muscle cells in the fore-stomach and hind-stomach. HGF was localized in smooth muscle cells (black arrows in Figure 3; white arrows in Appendix A), whereas weak HGF staining was seen in epithelial cells in the fore-stomach (black arrowheads in Figure 3; white arrowheads in Appendix A). MET expression was localized in epithelial cells of the fore-stomach (red arrowheads in Figure 3 and Appendix A), while strong MET expression was seen in the basal region of developing glandular structures in the hind-stomach (red arrowheads in Figure 3 and Appendix A).

We next analyzed the localization of tcHGF and tyrosine-phosphorylated MET (pMET) in day 16.5 embryos (Figure 4). In the fore-stomach and hind-stomach, HGF (scHGF + tcHGF) was mainly localized in smooth muscle cells, while tcHGF was diffusely localized in sub-epithelial cells (black arrows) and epithelial cells (yellow arrows). In the hind-stomach, strong tcHGF staining was seen in the base region of glandular structures (red arrow). These results suggest that some of the scHGF derived from smooth muscle cells was processed into tcHGF in sub-epithelial and epithelial cells. In the fore-stomach, strong MET staining was mainly seen in epithelial cells (red arrowheads), while strong MET expression was seen in the base region of developing glandular structures (red arrowheads) in the hind-stomach. pMET was localized in epithelial cells of the fore-stomach and hind-stomach (yellow arrowheads). Therefore, generation and localization of tcHGF may have played a role in MET activation in the developing stomach.

### 2.4. tcHGF and Phosphorylated MET Receptor in the Adult Stomach

To determine the potential involvement of HGF in gastric homeostasis, the localizations of tcHGF and pMET were analyzed in the stomach of 6–8-week-old hHGF-ki mice. In the adult stomach, the glandular epithelial structure is organized into corpus and antral regions. In the corpus, tcHGF was mainly localized in the glandular base region (Figure 5, red arrow), and extended to the neck region (yellow arrows), whereas HGF-positive cells were mainly seen in smooth muscle cells (black arrow). MET expression was present alongside glandular epithelial structures (black arrowheads). pMET was distributed in two distinct regions: the base region (red arrowhead), where chief and stem cells were located [25], and the isthmus region and/or the surface region (yellow arrowheads). Thus, tcHGF and pMET showed co-localization in the base region, which was also confirmed using double immunofluorescence staining (Appendix A). Substantially similar localization patterns were seen in the antrum (Appendix A).

Leucine-rich, repeat-containing G-protein coupled receptor 5 (Lgr5) expression is a definitive marker for gastric epithelial stem cells [25]. Since tcHGF and/or pMET showed close localization to gastric stem cells, we performed an immunostaining analysis in Lgr5-reporter mice in which the diphtheria toxin receptor-EGFP cassette was knocked into the endogenous Lgr5 locus [25]. Consistent with previous reports [25], GFP expression defined the localization of gastric stem cells at the base region of the corpus (Figure 6A,B) and antrum (Appendix A). Similar to the results in human HGF knock-in mice, pMET was distributed in two distinct regions: the base region (red arrowheads) and the isthmus and/or surface regions (yellow arrowheads) (Figure 6A). Double immunofluorescence staining also indicated the co-localization of pMET and GFP (white arrows) (Figure 6B); the numbers of GFP- and pMET-positive cells per corpus gland were 18.2 ± 2.9 and 17.7 ± 2.7, respectively (Figure 6C). Co-localization of GFP and pMET in the base region was also seen in the antrum (Appendix A).

### 2.5. Promotion of Gastric Organoids

Because the localization of pMET in Lgr5^+^ gastric stem cells suggests the involvement of the HGF–MET pathway in the growth and/or maintenance of gastric stem cells, we employed an organoid culture that allows three-dimensional reconstruction using various tissue stem cells, including gastric corpus organoids [25]. Corpus glandular basal cells were prepared and subjected to a three-dimensional culture (Figure 7). In the control culture, organoid formation was scarcely seen (Figure 7A,B), and viable gastric stem cells were not maintained during culture (Figure 7C). However, in the presence of epidermal growth factor (EGF), a standard growth factor for gastric organoid culture, or HGF, typical organoid structures were seen. Quantitative analysis of the size distribution of organoids (Figure 7B) and viable cells (Figure 7C) indicated that HGF promoted organoid formation and viability of gastric stem cells in a manner comparable to that of EGF. Because Lgr5+ gastric stem cells are indispensable for organoid formation, MET activation plays a role in the growth and/or maintenance of gastric stem cells.

### 2.6. Changes in tcHGF and pMET Distributions Post-Injury

tcHGF and pMET were detected in the area of the stomach where stem cells were present, and HGF facilitated stomach organoid formation. These results encouraged us to assess whether the HGF–MET pathway plays a role in stomach regeneration. For this purpose, we used repetitive tamoxifen treatment using hHGF-ki mice and analyzed changes in tcHGF and pMET localization over 7 days (Figure 8). Tamoxifen treatment damages the corpus epithelium, and is associated with the loss of parietal cells, proliferation of Lgr5^+^ gastric stem cells, and expansion of mucous neck cells [25,28,29]. On day 3 after the tamoxifen treatment, proliferating cells were seen in the glandular base region (Figure 8, black arrow) where strong pMET staining was detected (red arrowheads). pMET was also seen in the neck and/or surface regions (yellow arrowheads) where the expansion of mucous neck cells occurred [28,29]. tcHGF staining was strong in the base region (red arrows) and diffuse in the neck and/or surface regions (yellow arrows). On day 7, proliferating cells were mainly localized in the isthmus region where the recovery of parietal cells occurred. Strong staining of pMET and tcHGF was seen in the base region (red arrowheads and red arrows, respectively); pMET positivity was still present in the isthmus and/or surface regions. The proliferation of Lgr5^+^ gastric stem cells has been reported to occur 2 days after tamoxifen treatment and Lgr5^+^ cells have been shown to participate in subsequent epithelial regeneration [25]. Collectively, the localizations of pMET and tcHGF in glandular structures and their changes post-injury suggest that the HGF–MET pathway played a role in the homeostasis and regeneration of the stomach.

## 3. Discussion

A variety of bioactive molecules are secreted or stored as inactive forms and activated in response to developmental programs, injury, and/or pathology. Proteolytic conversion of inactive scHGF to active tcHGF has a key role in the activation of HGF–MET signaling; however, because of the lack of a molecular tool able to selectively capture tcHGF, the spatial and temporal regulation of tcHGF generation in development and tissue repair has remained unknown. In this study, we observed a change in the localization of tcHGF during gastric development, homeostasis, and regeneration by using a human tcHGF-specific monoclonal antibody and hHGF-ki mice. tcHGF was localized in areas that differ both from the primary site of synthesis and the localization of scHGF. Conversely, tcHGF co-localized with activated MET.

Previous studies suggested that HGF is a mesenchymal-derived factor that plays a role in epithelial cell growth, movement, and branched tube formation. In the developing stomach, we found that scHGF was mainly localized in smooth muscle cells, whereas tcHGF was diffusely distributed in sub-epithelial cells and also localized in epithelial cells. Our finding that the localization of tcHGF but not scHGF was closely related to pMET localization agrees with previous studies. Our work also indicated that the conversion of scHGF into tcHGF occurred in sub-epithelial and epithelial cells and that tcHGF acted on MET-expressing epithelial cells. Because glycosaminoglycans and collagens serve as binding sites for HGF in the extracellular matrix [30,31], scHGF was moderately diffusible, and was converted to tcHGF in the pericellular microenvironment.

The proteolytic processing of scHGF to tcHGF is catalyzed by several serine proteases [32]. Matriptase, a membrane-type serine protease, is broadly expressed in a variety of epithelial cells including human gastric epithelia [33,34]. The ablation of matriptase resulted in the impaired integrity and barrier function of epithelial tissues in the large intestine and epidermis [35,36]. HGF-activator is synthesized mainly in the liver and at a lower level in gastrointestinal tissue [32,37,38]. HGF-activator present in the plasma is activated by thrombin in injured tissues. Mice deficient in HGF-activator showed no abnormal tissue homeostasis, but the injured mucosa was not sufficiently covered by regenerated epithelium following colitis-induced injury [39]. A recent study reported that HGF-activator induced Go-to-G_Alert_ cell-cycle transition in stem cells in various tissues [40]. Taking these findings together, matriptase and HGF-activator are potential serine proteases responsible for tcHGF generation in the stomach.

Lgr5^+^ stem cells in gastrointestinal organs play important roles in homeostasis, regeneration, and cancer development [41]. The Lgr5^+^ stem cell-specific disruption of *MET* indicated that the MET receptor is dispensable for the normal homeostasis of intestinal epithelial cells [20]. However, MET deficiency in intestinal stem cells mildly attenuated intestinal regeneration after radiation-induced injury, presumably by impairing stem cell fitness [20]. The involvement of the HGF–MET pathway in regeneration was also supported by intestinal crypt organoid cultures, where HGF–MET signaling induced the outgrowth of intestinal stem cells into mini-guts, with an equivalent potency to EGF [20]. In addition, tumor microenvironment-derived HGF augmented Wnt/β-catenin signaling and clonogenicity in colon cancer stem cells characterized by high Wnt reporter activity [42]. Our study revealed that pMET and tcHGF were localized in the base region of the gastric corpus gland, and MET activation occurred in Lgr5^+^ gastric stem cells. Moreover, HGF induced organoid formation by gastric stem cells at a level comparable to that of EGF. Taken together, these findings show that although HGF is dispensable for the normal homeostasis of gastric stem cells, potentially because of redundant growth and survival signals, the HGF-MET pathway plays a supportive role in stem cell fitness, growth, and maintenance, and in stem cell-derived epithelial tissue expansion.

The HGF–MET pathway participates in homeostasis and regeneration in a variety of tissues. Studies of conditional MET-deficient mice have revealed that MET activation is important for the regeneration of organs such as the liver and skin [17,18,19], although the extent of its contribution depends on the tissue type. Ligand-dependent MET activation is closely associated with the acquisition of resistance to anticancer drugs in the cancer microenvironment [22]. Elucidation of the spatial and temporal patterns of tcHGF generation should promote our understanding of the precise mechanisms involved in tissue regeneration, stem cell activation, and cancer progression. To this end, molecular tools that selectively capture tcHGF should provide experimental opportunities to investigate these mechanisms.

## 4. Materials and Methods

### 4.1. Animal Experiments

All mouse experiments were performed in accordance with the Guide for the Care and Use of Laboratory Animals and were approved No. AD-173859 by the ethical committee of the Cancer Research Institute of Kanazawa University, Kanazawa, Japan. hHGF-ki mice were purchased from the Jackson Laboratory Office of Development (stock number: 014553, NSG-hHGFki; Bar Harbor, ME, USA). C57BL/6 mice were purchased from Sankyo Labo Service (Tokyo, Japan). Male hHGFki and C57BL/6 mice (8–10 weeks old) were maintained at the Advanced Science Research Center Institute for Experimental Animals, Kanazawa University. The fetuses were isolated from the uterus and dissected free of embryonic membranes in PBS. Tamoxifen (Merck, Kenilworth, NJ, USA) was dissolved in sunflower seed oil and injected intraperitoneally into 6–8-week-old mice at 4 mg/20 g body weight for 3 days. Tissues were prepared at 1, 3, and 7 days after treatment. Lgr5-DTR-EGFP mice were prepared as described previously [25].

### 4.2. Recombinant HGF

Full-length human HGF cDNA (NM_001010932.2) was used in all plasmid constructions throughout this study. Residue numbering was based on the sequence of variant 1, which contained five additional amino acids in the K1 domain. HGF cDNA with or without point mutations to eliminate N-linked glycosylation sites was as previously described [43]. Recombinant human HGF was expressed in Chinese hamster ovary cells, and secreted proteins were purified on an AKTA Purifier system (GE Healthcare Life Sciences, Pittsburgh, PA, USA), using a HiTrap Heparin HP Column (GE Healthcare) followed by size-exclusion chromatography on a Superdex 200 10/300 GL column (GE Healthcare) equilibrated in 20 mM Tris-HCl (pH 7.5) and 150 mM NaCl. Recombinant HGF was mainly composed of tcHGF and the content of scHGF was less than 1%. Recombinant HGF (Xa), in which the cleavage site of wild-type HGF (KQLR/V) was mutated to the recognition sequence of Factor Xa (IEGR/V), was prepared as previously described [26]. Briefly, HGF (Xa) was appended with a hexahistidine tag at the C-terminus and expressed in Expi293F cells (Thermo Fisher Scientific, Waltham, MA, USA). Secreted proteins were purified on a Ni-nitrilotriacetic acid (NTA) agarose column (Qiagen, Hilden, Germany). The C-terminal His-tag was eliminated using overnight incubation with Tobacco Etch Virus (TEV) protease. To prepare tcHGF (Xa), the TEV-treated samples as described above were further digested with 6 μg/mL Factor Xa (Novagen, Darmstadt, Germany). Recombinant scHGF (Xa) and tcHGF (Xa) proteins were further purified on a HiTrap Heparin HP Column (GE Healthcare) followed by size-exclusion chromatography on a Superdex 75 10/300 GL column (GE Healthcare) equilibrated in 20 mM Tris-HCl (pH 7.5) and 150 mM NaCl. Murine HGF was purchased from PeproTech, Inc. (Rocky Hill, NJ, USA).

### 4.3. Western Blotting

Proteins were subjected to SDS-PAGE in non-reducing conditions and electroblotted onto polyvinylidene difluoride (PVDF) membranes. The membrane was incubated in Tris-buffered saline containing 50 mM Tris-HCl, 150 mM NaCl, 0.1% Tween-20, and 3% (*w/v*) bovine serum albumin for 1 h at room temperature, and then treated with the following primary antibodies (diluted 1:1000): rabbit polyclonal mouse HGF antibody (LS-C294476; LifeSpan BioSciences, Inc., Seattle, WA, USA), mouse monoclonal anti-human HGF antibody (t5A11), or mouse monoclonal anti-human HGF antibody (t8E4). Monoclonal antibodies were prepared as described previously [26]. HRP-conjugated anti-rabbit IgG or anti-mouse IgG (Dako, Carpinteria, CA, USA) was used at 1:1000 dilution. Chemiluminescence was visualized and quantified using an ImmunoStar LD (Fujifilm Wako Pure Chemical, Osaka, Japan).

### 4.4. Immunohistochemical Staining

Tissues were fixed for 16–24 h in 4% paraformaldehyde in PBS at 4 °C. Tissues were dehydrated through a series of ethanol concentrations, immersed in xylene, and embedded in paraffin. Tissue sections at 5 µm were deparaffinized and rehydrated. Antigen retrieval was carried out by heating slides in a microwave oven for 10 min in 10 mM Tris-HCl buffer (pH 9.0) containing 1 mM EDTA. Endogenous peroxidase was inactivated via incubation in 3% hydrogen peroxide in PBS for 5–30 min. Sections were then blocked with PBS containing 3% bovine serum albumin for 30 min at room temperature. The following primary antibodies were used: anti-phospho MET (phospho Y1230 + Y1234 + Y1235) antibody (1:200 dilution, ab5662; Abcam), anti-proliferating cell nuclear antigen (1:4000, M0879; Dako, Glostrup, Denmark), anti-α-SMA antibody (at 1:200 dilution, ab5694; Abcam, Cambridge, UK), anti-MET (SP260)-R antibody (1:50 dilution, Sc-162-R; Santa Cruz Biotechnology), anti-GFP antibody biotin-conjugated (at 1:2000 dilution, 600-106-215; Rockland), anti-human HGF monoclonal antibody (t5A11) (1:250 dilution), and anti-human HGF monoclonal antibody (t8E4) (1:500 dilution). For immunohistochemical staining, tissue sections were visualized using a horseradish peroxidase (HRP)-labeled polymer system (Nichirei Bioscience, Tokyo, Japan) and DAB substrate (SK-4100; Vector Laboratories, Burlingame, CA, USA). For immunofluorescent staining, tissue sections were visualized using Alexa 488/594-labeled goat anti-mouse/rabbit antibodies or Alexa 488-labeled donkey anti-goat antibody (1:200 dilution; Thermo Fisher Scientific). Tissue sections were photographed using an Olympus BX51 microscope (Olympus, Tokyo, Japan) or Biozero BZ-9000 (Keyence, Osaka, Japan). The negative controls for immunohistochemical staining were performed without using primary antibodies. HRP-labeled anti-rabbit IgG or anti-mouse IgG was used as the secondary antibody (Appendix A).

### 4.5. Organoid Culture and Viable Cell Assay

Isolation of the murine corpus and organoid culture were performed as described previously [25]. Briefly, a corpus isolated from a C57BL/6 mouse was incubated in chelation buffer and glands were isolated via repeated pipetting, filtration, and centrifugation. Glands were digested to single cells in TrypLE (Life Technologies, Carlsbad, CA, USA) with DNaseI (0.8 U μL^−1^) (Sigma, Darmstadt, Germany), and single corpus cells isolated using fluorescence-activated cell sorting were seeded in Matrigel and cultured in medium supplemented with EGF, gastrin, fibroblast growth factor-10 (FGF10), 50% L-cell line to secrete Wnt3a, R-spondin 3 and Noggin (L-WRN) conditioned medium containing noggin, Wnt3a, and R-spondin, and Rock-inhibitor (Y27632). Organoid cultures were passaged every 3 days. To test the effect of HGF on the growth of organoids, L-WRN conditioned medium was reduced to 10%, and EGF and FGF10 were omitted from the culture medium. Cell viability and growth in the 3D organoid culture were measured using the CellTiter-Glo^®^ 3D cell viability assay kit (Promega, Madison, WI, USA), in accordance with the manufacturer’s instructions. The luminescence signal was measured at 560 nm.

## Figures and Tables

**Figure 1 ijms-20-02955-f001:**
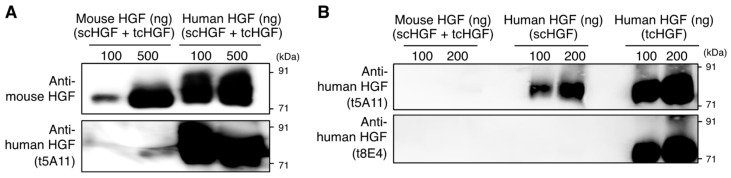
Specificity of anti-human HGF antibodies. (**A**) Western blot analysis for the cross-reactivity of anti-mouse HGF polyclonal antibody and t5A11 anti-human HGF monoclonal antibody between mouse and human HGF. (**B**) Specificity of anti-HGF monoclonal antibodies (t5A11 and t8E4) to mouse HGF, scHGF, and tcHGF. Mouse HGF and human HGF were subjected to SDS-PAGE under non-reducing conditions. Human HGF (composed of scHGF and tcHGF), scHGF, and tcHGF were expressed and purified as described in the Materials and Methods section.

**Figure 2 ijms-20-02955-f002:**
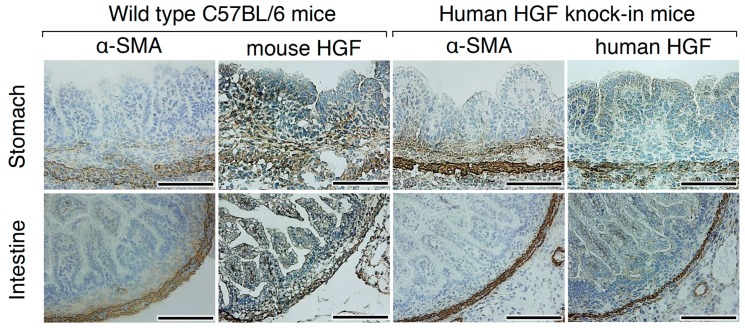
Localization of HGF in the developing stomach and intestine of wild-type C57BL/6 and hHGF-ki mice. Immunohistochemistry was performed using anti-mouse HGF polyclonal antibodies and t5A11 anti-human HGF monoclonal antibodies, respectively, in C57BL/6 and hHGF-ki mice. Similar expression and localization patterns were obtained in sections from two different mice. Tissues were obtained from day 16.5 embryos. Scale bars represent 200 µm.

**Figure 3 ijms-20-02955-f003:**
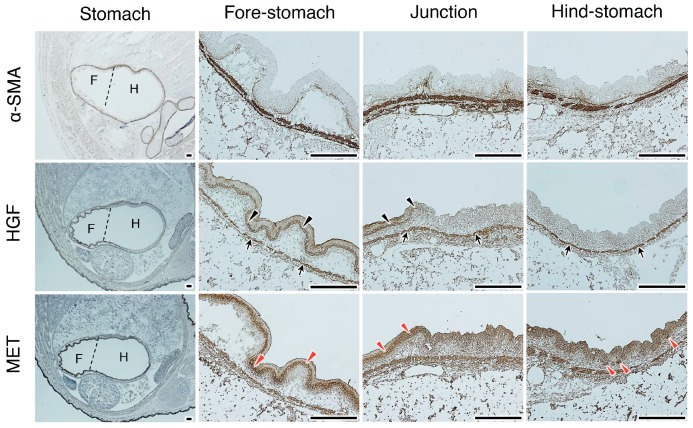
Localizations of HGF and MET receptors in the developing stomach. Immunohistochemical staining was performed using t5A11 anti-human HGF monoclonal antibody or anti-MET antibody. Stomachs in day 16.5 embryos were divided into the anterior/fore-stomach and posterior/hind-stomach, distinguished by dotted lines. Black arrows indicate HGF localization in smooth muscle cells. Black arrowheads indicate HGF localized in the epithelial cells. Red arrowheads indicate MET expression in epithelial cells. Similar localization patterns were obtained in sections from two different mice. Tissues were obtained from day 16.5 embryos of hHGF-ki mice. Scale bars represent 200 µm.

**Figure 4 ijms-20-02955-f004:**
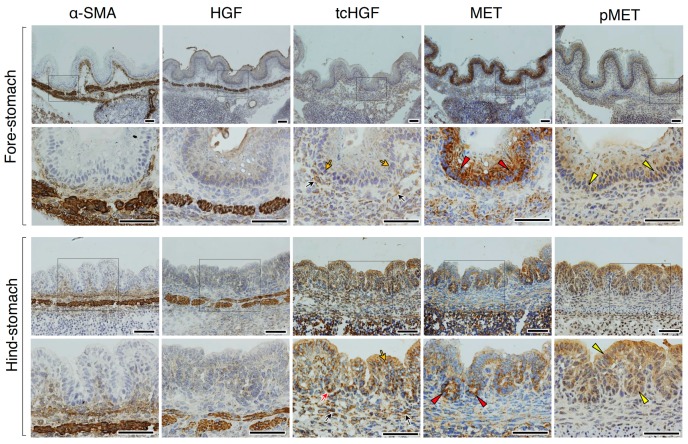
Localizations of HGF, tcHGF, MET, and phosphorylated MET (pMET) in the developing stomach. Immunohistochemical staining was performed using t5A11 (for scHGF and tcHGF), t8E4 (for tcHGF), anti-MET, or anti-phospho-MET antibody. The fore-stomach and hind-stomach were respectively characterized by multilayered squamous epithelia and developing glandular epithelium. Black and yellow arrows indicate tcHGF localized in the sub-epithelial and epithelial cells, respectively. The red arrow indicates strong tcHGF localization in developing glandular regions. Red arrowheads indicate strongly MET-positive cells in the epithelial cells. Yellow arrowheads indicate pMET in epithelial cells. The images in the lower panel are magnified images of the boxed areas in the upper panel. Similar immunohistochemical localization patterns were obtained in sections from two different mice. Tissues were obtained from day 16.5 embryos of hHGF-ki mice. Scale bars represent 200 µm.

**Figure 5 ijms-20-02955-f005:**
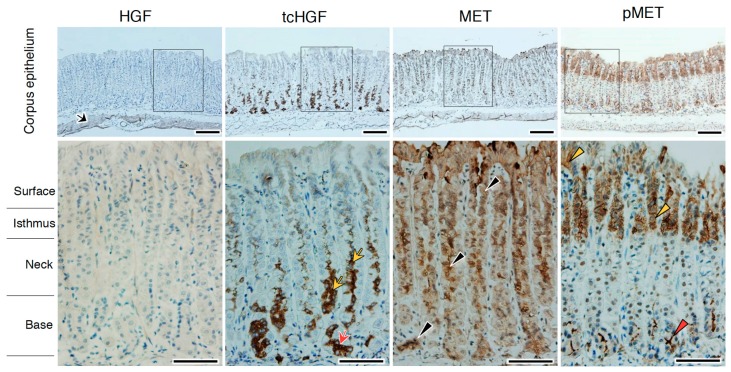
Localizations of HGF, tcHGF, MET, and pMET in the corpus epithelium of the adult stomach. Immunohistochemical staining was performed using t5A11 (for scHGF and tcHGF), t8E4 (for tcHGF), anti-MET, or anti-phospho-MET antibody. Black arrows, HGF in smooth muscle cells; red arrow, tcHGF in the glandular base region; yellow arrows, tcHGF in the neck region; black arrowheads, MET in glandular epithelial structure; red arrowhead, pMET in the glandular base region; yellow arrowhead, pMET in the isthmus region and the surface region. The images in the lower panel are magnified images of the boxed areas in the upper panel. Similar immunohistochemical localization patterns were obtained in sections from two different mice. Tissues were obtained from hHGF-ki mice. Scale bars represent 200 µm.

**Figure 6 ijms-20-02955-f006:**
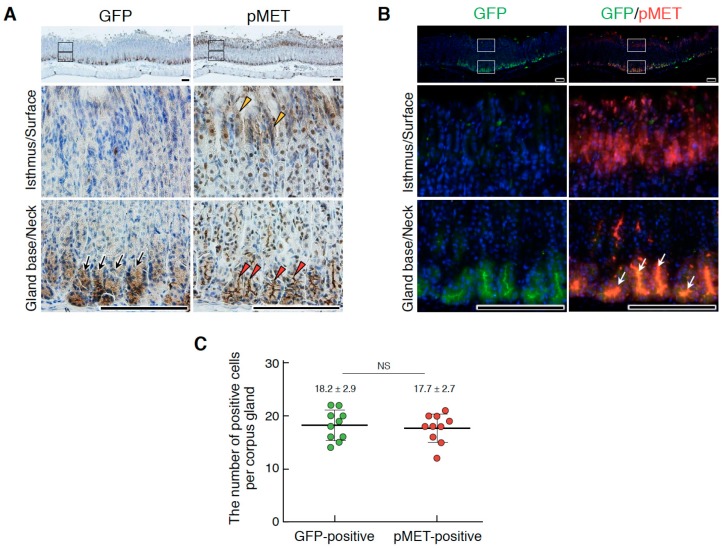
Localizations of Lgr5^+^ stem cells and pMET in the corpus epithelium of the adult stomach. (**A**) Immunohistochemical staining for GFP and pMET. (**B**) Double immunofluorescence staining for GFP and pMET. In (**A**), black arrows, Lgr5-driven GFP; red arrowheads, pMET in the glandular base region; yellow arrowheads, pMET in the isthmus and/or surface regions. In (**B**), white arrows indicate the co-localization of pMET and GFP in Lgr5^+^ stem cells. The images in the lower panel are magnified images of the boxed areas in the upper panel. Tissues were obtained from Lgr5-DTR-EGFP mice. Scale bars represent 200 µm. (**C**) The numbers of GFP- and pMET-positive cells per corpus gland. These values were obtained by counting in 10 glands (*n* = 10). Data were tested for significance using an unpaired two-tailed *t*-test. NS, not significant.

**Figure 7 ijms-20-02955-f007:**
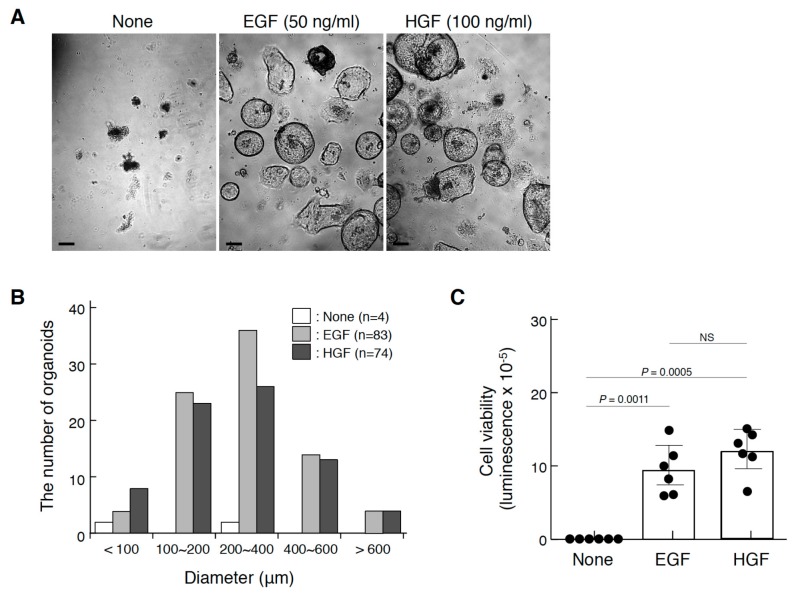
Promotion of gastric corpus organoids by HGF. (**A**) The appearance of organoids cultured in the absence or presence of EGF or HGF. Scale bars represent 200 µm. (**B**) The size distribution of organoids. (**C**) The viability of organoids. In (**A**), corpus gland stem cells isolated from a C57BL/6 mouse were cultured in Matrigel in the absence or presence of EGF (50 ng/mL) or recombinant HGF (100 ng/mL) for 21 days. In (**B**), the size distribution was analyzed by using Image-J. In (**C**), the viability was measured using the CellTiter-Glo^®^ 3D cell viability assay. Values are presented as mean ± S.D. (*n* = 6). Data were tested for significance using an unpaired two-tailed *t*-test. NS, not significant. Organoid culture was performed twice, from which similar results were obtained.

**Figure 8 ijms-20-02955-f008:**
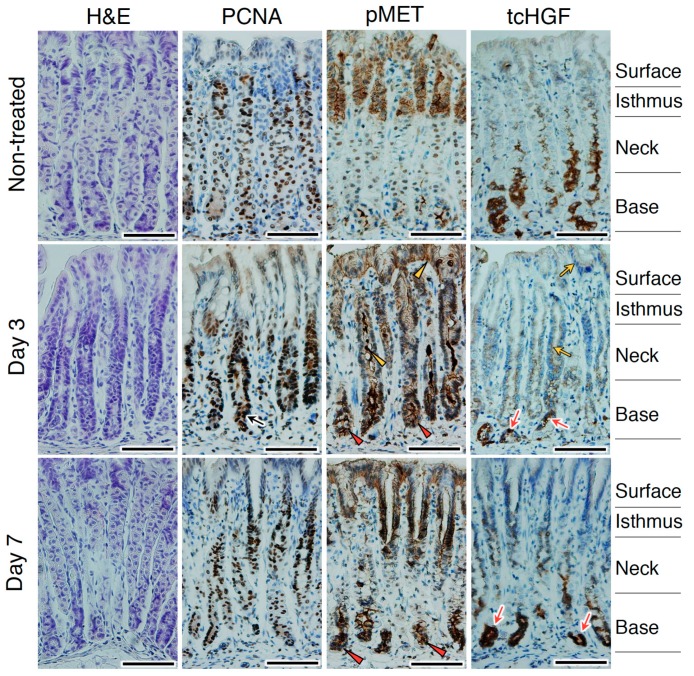
Changes in the localizations of tcHGF and pMET following stomach injury. hHGF-ki mice were subjected to a tamoxifen treatment. The black arrow indicates PCNA-positive proliferating cells. Red arrowheads and red arrows indicate pMET and tcHGF localizations, respectively, in the glandular base region. Yellow arrowheads and yellow arrows indicate pMET and tcHGF localizations, respectively, in the neck and/or surface regions. H&E, hematoxylin and eosin. Similar results were obtained in sections from two different mice independently subjected to tamoxifen treatment. Scale bars represent 200 µm.

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
