# Peer review of "Distinct Localization of Mature HGF from its Precursor Form in Developing and Repairing the Stomach"

_ijms, 2019, doi:10.3390/ijms20122955_

Round 1

Reviewer 1 Report

The authors performed all the experiments requested.

-  The ELISA binding assay has been performed and the data inserted in Figure S2

-  Co-staining of α-SMA and HGF has been performed and the results inserted in Figure S3.

-  Co-staining of tcHGF and pMET has been performed and the results inserted in Figure S6.

-  The activity of EGF and HGF on the viability of gastric organoids has been investigated and the results are reported in Figure 7C. A quantification of the size distribution of organoids has also been inserted.

The text has been revised, but amendments are still necessary to clarify some points of the text. The words or sentences that need to be modified have been highlighted within the manuscript and suggestions inserted. 

Author Response

Response to Reviewer-1’s Comments

The reviewer’s comment:

The authors performed all the experiments requested.

-  The ELISA binding assay has been performed and the data inserted in Figure S2

-  Co-staining of α-SMA and HGF has been performed and the results inserted in Figure S3.

-  Co-staining of tcHGF and pMET has been performed and the results inserted in Figure S6.

-  The activity of EGF and HGF on the viability of gastric organoids has been investigated and the results are reported in Figure 7C. A quantification of the size distribution of organoids has also been inserted.

The text has been revised, but amendments are still necessary to clarify some points of the text. The words or sentences that need to be modified have been highlighted within the manuscript and suggestions inserted. 

Response:

We deeply appreciate your helpful and kind suggestions for our manuscript. Our manuscript has been corrected completely according to your suggestions in the text. We wish to express our sincere thanks for your careful review and suggestions.

Reviewer 2 Report

The authors have addressed the issues that were raised and improved the manuscript significantly.  

Author Response

Response to Reviewer-2’s Comments

The reviewer’s comment:

The authors have addressed the issues that were raised and improved the manuscript significantly.

Response:

We wish to express our sincere thanks for your careful review and comments.

This manuscript is a resubmission of an earlier submission. The following is a list of the peer review reports and author responses from that submission.

Round 1

Reviewer 1 Report

The authors investigate the localization of mature two-chains HGF (tcHGF) in the developing and regenerating stomach. To this aim, they exploit a panel of antibodies previously developed by their group. In particular, the t5A11 antibody recognizes both scHGF and tcHGF, while the t8E4 antibody selectively binds human tcHGF, thus allowing to discriminate between the single chain inactive precursor (scHGF), and the processed mature form (tcHGF) that triggers MET activation. The authors also take advantage of a transgenic mouse model (human HGF knock in mice) that  allows to detect the distribution of human HGF in murine organs, thanks to the use of antibodies specifically recognizing human HGF.

General comment

The paper is largely descriptive, based on a meticulous immunohistochemical analysis of the expression of tcHGF in correlation with the expression and/or phosphorylation of its receptor MET. Only in one figure (n. 6) and in one Supplementary Figure (n. 3) the authors perform an immunofluorescence analysis, that allows the co-localization of different signals. The language needs a thorough revision: the sentences are often confused due to the presence of grammatical and/or syntactical errors that in some points make it difficult to understand the text. Some figure legends also need to be revised.

Detailed comments

1 – Results 2.1. lines 94-104  The t5A11 and t8E4 monoclonal antibodies (mabs) have been thoroughly characterized previously (ref. 26) as far as their selectivity for scHGF and tcHGF is concerned. In the same paper, however, their species-specificity is not mentioned. In this study, the authors evaluate the ability of the two mabs to recognize mouse and/or human HGF by western blotting. Since this is not a sensitive technique,  ELISA binding assays with increasing concentrations of mouse and human HGF should be performed.

 2 – Results 2.2. lines 125-131 and Figure 2 To support the coexistence of HGF and α-SMA, the authors should provide an immunofluorescence analysis of the stomach of wild type and hHGF knock in mice showing co-localization or proximity of the two signals. The whole paragraph is confused and should be re-written. In the panel of stomach and intestine immunohisto-chemistry, the images in the lower panel are a magnification of the corresponding upper image: this should be stated in the figure legend (the same is true for legends to figure 4 and 5).

3 – Figure 3 The legend to the figure should be revised.

4 – Results 2.4. lines 224-226  To sustain the overlapping of tcHGF and pMET distribution, the authors should provide an immunofluorescence analysis showing co-localization of the two signals.

5 – Results 2.5. To state that HGF and EGF have comparable effects on the formation and growth of gastric organoids, the authors should provide data on the biological activities of the two molecules. Viability  or proliferation assays on organoids should be performed.

6 – Results 2.6. The whole paragraph is confused and should be re-written.

7 - Discussion  This section also needs a grammatical/syntactical revision. The eventuality that scHGF to tcHGF processing may support organ regeneration by participating in the Go-to GAlert cell cycle transition is fascinating, but it is pure speculation and should be substantiated by some preliminary experimental observations.

Reviewer 2 Report

In the current study, the authors examine the role of hepatocyte growth factor, HGF in development and injury repair in the stomach.  Using a novel monoclonal antibody that specifically detects the activated form of HGF (tcHGF), a series of immunohistochemical experiments aimed at determining the localization of tcHGF in relation to the activated pMET receptor were performed in stomach and other tissues, stem cell compartment, and regenerating stomach tissue after treatment with tamoxifen, as well as in organoid cultures.  Their studies showed that tcHGF and pMET altered their tissue localization and showed colocalalization in response to gastric  injury and in stem cell rich regions. In vitro, HGF promoted the formation of organoids from gastric stem cells to a comparable level as EGF.  Taken together, they conclude that activated HGF interacts with pMET to play an important role in organ development as well as in tissue repair in response to injury in a mechanism that is mediated by the microenvironment.

In general, the conclusions were supported by the data presented in the study and the manuscript was well written.  It would have been more impactful if the colocalization experiments were done by immunofluorescence using differentially labeled antibodies similar to that shown in Supplemental Figure 3.  The use of  immunohistochemistry made it difficult to delineate the positive staining for the molecules indicated in each of the figures. A few points need to be addressed as indicated below:

1) On page 4, last sentence of the first paragraph:  Unpublished date regarding the detection of human HGF but not mouse HGF in plasma of transgenic mice should be presented as a supplemental Figure. 

2)  In Figure 7, was full length HGF used to promote development of organoids?  Did the organoids express high levels of pMET or tcHGF?  Was interaction between HGF and MET required for enhanced organoid development?

While most of the conclusions were based on colocalization of tcHGF and MET, this experiment provides a functional assay to show that potential interactions between these molecules are critical for organoid development. The additional data should be provided to show that these interactions provide the underlying mechanism for enhanced development of organoids.

3)  In supplemental Figure 3, the Figure legend states that the anterior/fore-stomach and posterior/hind-stomach regions are distinguished by "dotted lines" , however, the dotted lines are not clearly visible in the figures.  

Reviewer 3 Report

In this article the authors Jangphattananont et al. study localization of proHGF (a single chain HGF), active HGF (two chain HGF), MET and activated, phospho-MET, during development and during injury in the stomach.  They have generated a unique antibody that specifically recognizes the active (two chain) HGF.

The authors find that precursor and active HGF have distinct localization during development and that HGF is activated during injury-induced repair in the stomach.

Major issues:

1.  The authors find that during development, HGF is restricted to a-SMA positive cells, which they refer to as smooth muscle cells.  While aSMA is indeed expressed in smooth muscle cells, it also marks myofibroblasts, which can produce HGF.  How can the authors be sure that smooth muscle cells and not the myofibroblasts produce pro-HGF?

2.   The quality of immunohistochemistry images should  be improved;  the images should be shown at a higher magnification-  a part of each image could be shown as an inset with higher magnification.  This would allow the reader to conclude about the subcellular localization of MET/pMET and HGF.  

     It is not  clear at all where do arrows point to on some figures (see for example Fig. 4 and Fig 5; as indicated, the yellow and the red arrows point often to empty spaces).  Better quality images with higher magnification are needed for immunohistochemistry data, and negative controls (no primary antibody) for staining should be shown in Supplemental data (at least for some crucial experiments).

3.   Processing of pro-HGF is mediated – mainly- by three serine proteases, matriptase, hepsin and HGFA.  What is known about the location and/or the activity of these enzymes in the stomach during development and during injury?

4.   Data presented in Fig. 8 are not very convincing, larger magnification images are required.   

5.   The manuscript would be strengthened if some of the images are quantified. For example, in Fig. 6, the number of GFP and pMET positive cells per crypt could be calculated.

Minor issues:

1. Supplemental Fig. 1 needs a figure legend.